# Modulation of the PGE_2_-Mediated Pathway in the Eclosion Blocking Effect of Flumethrin and Terpenoid Subfraction Isolated from *Artemesia nilagirica* in *Rhipicephalus annulatus*

**DOI:** 10.3390/molecules26164905

**Published:** 2021-08-13

**Authors:** Panicker Devyani Ramachandran, Mahesh Doddadasarahalli Muniyappa, Sreelekha Kanapadinchareveetil, Suresh Narayanan Nair, Karapparambu Gopalan Ajithkumar, Sujith Samraj, Anoopraj Rajappan, Anju Varghese, Deepa Chundayil Kalarickal, Reghu Ravindran, Srikanta Ghosh, Sanis Juliet

**Affiliations:** 1Department of Veterinary Pharmacology and Toxicology, College of Veterinary and Animal Sciences, Pookode, Kerala Veterinary and Animal Sciences University, Lakkidi, P. O., Wayanad 673576, Kerala, India; devyanip1512@gmail.com (P.D.R.); maheshdm1132@gmail.com (M.D.M.); sreelekha.kp@gmail.com (S.K.); suresh@kvasu.ac.in (S.N.N.); sujith@kvasu.ac.in (S.S.); sanis@kvasu.ac.in (S.J.); 2Department of Veterinary Parasitology, College of Veterinary and Animal Sciences, Pookode, Kerala Veterinary and Animal Sciences University, Lakkidi, P. O., Wayanad 673576, Kerala, India; ajithkg@kvasu.ac.in (K.G.A.); anju@kvasu.ac.in (A.V.); deepack@kvasu.ac.in (D.C.K.); 3Department of Veterinary Pathology, College of Veterinary and Animal Sciences, Pookode, Kerala Veterinary and Animal Sciences University, Lakkidi, P. O., Wayanad 673576, Kerala, India; anoopraj@kvasu.ac.in; 4Division of Parasitology, ICAR-Indian Veterinary Research Institute, Izatnagar 243122, Bareilly, India; sghoshtick@gmail.com; 5Center for Ethnopharmacology, College of Veterinary and Animal Sciences, Pookode, Kerala Veterinary and Animal Sciences University, Lakkidi, P. O., Wayanad 673576, Kerala, India

**Keywords:** adult immersion test, LC-MSMS analysis, tick, ovary, immunohistochemistry, pyrethroid

## Abstract

Prostaglandins are a group of important cell-signaling molecules involved in the regulation of ovarian maturation, oocyte development, egg laying and associated behaviors in invertebrates. However, the presence of prostaglandin E_2_ (PGE_2_), the key enzymes for PGE_2_ biosynthesis and its interference by drugs were not investigated previously in the ovary of ticks. The present study was undertaken to assess the modulation of the PGE_2_-mediated pathway in the eclosion blocking effect of flumethrin and terpenoid subfraction isolated from *Artemisia nilagirica* in *Rhipicephalus annulatus* ticks. The acaricidal activities and chemical profiling of the terpenoid subfraction were performed. The localization of the cyclooxygenase1 (COX1) and prostaglandin E synthase (PGES) enzymes and the quantification of PGE_2_ in the ovaries of the ticks treated with methanol (control), flumethrin and terpenoid subfraction were also undertaken. In addition, the vitellogenin concentration in hemolymph was also assayed. Both flumethrin and the terpenoid subfraction of *A. nilagirica* elicited a concentration-dependent inhibition of fecundity and blocking of hatching of the eggs. The COX1 could not be detected in the ovaries of treated and control ticks, while there was no significant difference observed in the concentration of vitellogenin (Vg) in them. The presence of PGES in the oocytes of control ticks was confirmed while the immunoreactivities against PGES were absent in the vitellogenic oocytes of ticks treated with flumethrin and terpenoid subfraction. The levels of PGE_2_ were below the detection limit in the ovaries of the flumethrin-treated ticks, while it was significantly lower in the ovaries of the terpenoid subfraction-treated ticks. Hence, the prostaglandin E synthase and PGE_2_ were identified as very important mediators for the signaling pathway for ovarian maturation and oviposition in ticks. In addition, the key enzyme for prostaglandin biosynthesis, PGES and the receptors for PGE_2_ can be exploited as potential drug targets for tick control. The detection of PGES by immunohistochemistry and quantification of PGE_2_ by LC-MSMS can be employed as valuable tools for screening newer compounds for their eclosion blocking acaricidal effects.

## 1. Introduction

Ticks are obligate hematophagous ectoparasites distributed all over the world, transmitting a multitude of pathogens to both animals and humans, and causing a substantial economic burden. The occurrence of tick-borne diseases has increased in recent years, causing major health problems in animals as well as humans [1]. Nearly 904 tick species were reported throughout the world [2] and 109 from India [3]. The economically important tick species *Rhipicephalus microplus* is the predominant species in northern parts of India [3] while *R. annulatus* is considered as the commonest tick species of southern India [4,5].

Strategies laid down for sustainable tick control include chemotherapeutic, biological, herbal and immunological methods [6]. The application of synthetic chemical acaricides is the most practical and widely used control method worldwide. Commercially available synthetic acaricides such as organophosphates, halogenated hydrocarbons and pyrethroids have shown long environmental half-lives and also possess toxicity problems. Hence, there is an urgent need for alternate, environmentally benign, toxicologically safe, more selective and efficacious pesticides [7].

The role of secondary metabolites of plants such as phenols, flavonoids, terpenoids, alkaloids, essential oils, quinones, tannins, saponins and sterols in eliciting defense against pests is well known. The extracts of many plants with promising acaricidal properties were reported from our laboratory too [8,9,10,11,12,13,14,15,16,17,18]. In addition to their direct acaricidal activity due to neurotoxicity, many herbal extracts produced accelerated degeneration of salivary glandular tissue and changes in the morphophysiology of reproductive organs in ticks [19,20]. The histopathological and ultrastructural alterations induced by phytoacaricides [19,21,22,23,24,25,26,27,28,29,30,31,32] were comparable to those induced by conventional synthetic chemical acaricides [33,34,35,36,37,38,39,40,41].

*Artemisia nilagirica* (Clarke) Pamp., a tall aromatic herbaceous perennial shrub belonging to the family Asteraceae, locally known as “Indian wormwood”, has demonstrated antimalarial, anthelmintic, antifungal, anti-inflammatory and astringent activities [42,43]. Remarkable larvicidal, pupicidal, adulticidal as well as repellent properties against *Anopheles stephensi* and *Aedes aegypti* mosquitoes [44] and acaricidal potential against *R. annulatus* [45] of this plant extract were already reported. A recent review article on secondary metabolites of the genus *Artemisia* revealed the pesticidal activity of the mono- and sesquiterpenes of the plant [46].

In arthropods, numerous developmental and reproductive processes such as molting, growth, metamorphosis and gonad maturation are regulated by the major sesquiterpenoid hormones, juvenile hormone (JH) and methyl farnesoate (MF) [47]. In non-insect arthropods like crustaceans, MF was the major form of sesquiterpenoid hormone which regulates the gonadal maturation and induced vitellogenesis [47]. In ticks, the molting hormone ecdysteroid plays an important role in oogenesis and oviposition by triggering vitellogenesis [48]. In addition, prostaglandin E_2_ (PGE_2_), abundantly present in salivary secretions and the ovary of ticks, also plays important roles in peristaltic activity of oviduct to transport oocytes to genital aperture [49]. Earlier studies were mainly focused on the roles of prostaglandins at the host-tick interface [50]. The major roles of PGE_2_ in the regulation of ovarian maturation, oocyte development, egg-laying and associated behaviors of insects, fish, crustaceans and arthropods are well documented [51,52,53,54,55,56].

Cyclooxygenases (COX) and Prostaglandin G/H synthase (PGHS) are the key enzymes involved in the prostaglandin biosynthesis in all vertebrates and invertebrates. They are found abundantly in the nuclear envelope and endoplasmic reticulum of insects, as well as in crustaceans [55,57]. However, the available reports on the detection and the function of COX in the tick ovary are scanty.

Many terpenoids have a role in the modulation of reproduction. The sesquiterpene compound, cadina-4,10 (15)-dien-3-one, inhibited lipid metabolism during embryogenesis of *R. microplus* eggs in a dose-dependent manner [58]. Naturally occurring sesquiterpene lactones and their semi-synthetic derivatives cause a reduction in the levels of PGE_2_ in radiation-induced fibroma cells (RIF1) of the mouse by inhibiting COX2 activity and expression [59].

Previously, Udayan et al. [45] revealed the eclosion blocking activity of three subfractions from the active hexane fraction of the ethanolic extract of *A. nilagirica* on the eggs laid by the treated *R. annulatus* ticks. The incomplete maturation of oocytes observed could be due to the inhibition of biosynthesis, release or activity of PGE_2_, resulting in the inhibition of ecdysteroid synthesis and thereby the vitellogenesis process. Hence, the present study aims to assess the modulation of the PGE_2_-mediated pathway of terpenoid subfraction isolated from hexane extract of *A. nilagirica* in producing the eclosion blocking effect in *R. annulatus*.

## 2. Results and Discussion

### 2.1. Gas Chromatography–Mass Spectrometry Analysis of Terpenoid Subfraction

The analysis of the terpenoid subfraction isolated from the hexane fraction of the ethanolic extract of aerial parts of *A. nilagirica* by GC/MS revealed the presence of twenty-two terpenoids. Among them, the major compounds identified were isothujol (relative concentration = 14.6%), phytol (9%), neoisothujol (6.59%), β-eudesmol (4.27%), platambin (2.17%), nootkatone (1.71%), caryophyllene oxide (1.51%), 3-thujanone (0.84%), and alloaromadendrene (0.62%). The identified chemical components with their retention time and the area per cent are presented in Table 1.

The presence of a high content of sesquiterpenoids and cyclocolorenones in the hexane fraction of *A. nilagirica* was reported recently [45]. The presence of elevated content of the terpenoids in *A. nilagirica* collected from higher altitudes was also reported [60,61,62]. The chemical composition of plants, as well as their biological activity, may vary based on the geographical location as well as on the seasons and changes in the climate [63,64,65]. The changes in the composition were also reported during the blooming and fruiting stages [66]. The variation could also be noticed in the composition between the essential oil hydro-distilled from the aerial parts of *A. nilagirica* grown in northern hilly and southern hilly regions of the Indian subcontinent [43,67].

### 2.2. Adult Immersion Test

The results of the adult immersion test (AIT) using the terpenoid subfraction obtained from the hexane fraction of the ethanolic extract of the aerial parts of *A. nilagirica* against *R. annulatus* are given in Table 2. The terpenoid subfraction of *A. nilagirica* showed a concentration dependent effect on the hatching per cent of the laid eggs with the highest inhibition of 50% at 10% concentration. The per cent inhibition of fecundity and adult mortality were concentration-dependent. The LC_50_ value of the terpenoid subfraction of *A. nilagirica* against the tick *R. annulatus* was 28.46%. The dose-response curve based on the probit analysis is shown in Figure 1.

The acaricidal properties of the plant extracts against *R. microplus* were attributed mainly to the terpenoids [68,69,70,71]. Recently, the crude ethanolic extract and the hexane fraction of the ethanolic extract of *A. nilagirica* were reported with concentration-dependent larvicidal, and adulticidal activities against *R. annulatus* [45]. Furthermore, the study also showed the higher effectiveness of the hexane fraction on the inhibition of fecundity over ethanolic extract against *R. annulatus* ticks. Moreover, the terpenoid subfraction isolated from the hexane extract of *A. nilagirica* elicited 100% mortality of *R. annulatus* larvae [72].

### 2.3. Histology of Tick Ovary

The ovary of *R. annulatus* revealed the presence of more numbers of oocytes of stages III, IV and V. The attachment of oocytes to the germinal epithelium was by the pedicel. The stage I oocyte was small, elliptical and with the germinal vesicle present in the cytoplasm. The stage II oocytes were slightly larger than the first stage with germinal vesicle present centrally in the cytoplasm. The stage III oocytes were larger and rounded with the cytoplasm filled with small yolk droplets. The germinal vesicle of the stage III oocytes was observed towards the pole in contact with the pedicel. The stage IV oocytes were much larger when compared with stage III oocytes, and elliptical to round in shape. The large yolk droplets were fully formed throughout the cytoplasm and the germ vesicle was almost invisible. The stage V oocytes were the largest and are spherical with large yolk droplets merged in the center. The germ vesicle was invisible in them. The histological findings in the ovary of *R. annulatus* were similar to our previous report [73].

The histological sections of ovaries of ticks treated with flumethrin and terpenoid subfraction showed many oocytes with morphological changes compared to the control. In the ovary of the ticks treated with flumethrin, the cytological changes were observed in the pedicel. The oocytes of all stages revealed a drastic reduction in the size, irregular shape, disruption of cytoplasm, interrupted margins and degeneration of the nuclear region. The cell boundaries were irregular and the chorionic layer exhibited interruptions. A decrease in the content of the yolk droplets and an increase in the number of vacuoles in the stage IV and V oocytes were also observed.

Unlike flumethrin, the changes caused by the terpenoid subfraction were of a lower degree. The terpenoid subfraction did not interrupt the outer chorionic layer. Oocytes were deformed with the break in the nucleus region. The abnormal morphological and structural changes observed in the present study were previously reported in the ovaries of the ticks treated with other phytoacaricides [21,29] and synthetic acaricides like deltamethrin, amitraz [38,39,40], fipronil [33,34], permethrin [36,37] and cypermethrin [74]. The above studies indicated the inhibitory effect of these acaricides on the oocytes at different stages of their development. This inhibition might have caused the decrease in the number of eggs and their viability following treatment with terpenoid subfraction or flumethrin. The reproductive disruption thus caused could be mediated either through its direct effect on the reproductive tissue and/or indirectly through the inhibition of the endocrine process [75].

#### 2.3.1. Immunoperoxidase Staining of the Ovarian Sections with Anti-COX1 Antibodies

The immunoperoxidase staining of the ovary of ticks treated with methanol (control), flumethrin and terpenoid subfraction using anti-COX1 antibodies are depicted in Figure 2A–C respectively. The ovarian tissues of the ticks treated with methanol, flumethrin and terpenoid subfraction did not reveal the presence of the COX1 enzyme.

The role of COX1 and COX2 enzymes in the reproduction was previously explained using the knockout mouse model. The mice deficient in COX1 exhibited longer gestation periods, protracted parturition and delivery of few live young mice compared to wild mice. The conception and fetal development were unaltered in the mice deficient with COX1 indicating that the prostanoids synthesized by COX1 were not important for ovulation, fertilization or implantation, but were essential for parturition. However, when there was ablation of the COX2 gene in mice, multiple reproductive failures, ovulation, fertilization, implantation and decidualization were noted, confirming that the prostaglandins synthesized by COX2 played an important role in these reproductive processes. Thus, both COX isoforms catalyzed the same reaction but then differed in their distribution in the reproductive tract [76].

The synthesis of the prostaglandins occurred through membrane-derived arachidonic acid (AA) via three sequential enzymatic reactions involving phospholipase A_2_ enzymes, cyclooxygenases and specific terminal prostanoid synthases. The segregated utilization of COX1 and COX2 were observed in the prostaglandin biosynthesis pathway even though they were present together in the same cells. The constitutive COX1 enzyme was mainly seen in immediate prostaglandin synthesis, which occurred due to stimulation of Ca^2+^ mobilizers and was functionally coupled with cytosolic prostaglandin E synthase (cPGES). The inducible COX2 was mainly seen in delayed prostaglandin synthesis and was functionally coupled with microsomal prostaglandin E synthase-1 and 2 (mPGES-1 and mPGES-2). The COX1 was mainly localized in the endoplasmic reticulum and perinuclear membranes, whereas COX2 was located in the perinuclear envelope. The cPGES was located mainly in the cytosol and was expressed in a variety of cells and tissues as it was linked with constitutive COX1. The mPGES-1 located in the perinuclear membrane and functionally coupled with COX2 was mainly targeted by cancer drugs. The mPGES-2 localized in the Golgi apparatus lead to the maturation of cytosolic enzymes, was linked with both COX1 and COX2 [77]. The biosynthetic enzymes COX2 and mPGES were the two inducible enzymes that facilitated PGE_2_ synthesis in synergy. The COX1 enzymes needed higher concentrations of arachidonic acid for their proper functioning than COX-2 [78].

In the present study, COX1 could not be localized in the oocytes of the ovaries of *R. annulatus* ticks treated with methanol (control), flumethrin (90 ppm) and terpenoid subfraction (40 per cent). The section of these ovaries mainly revealed the presence of stage III, IV and V oocytes. The stage III oocytes were large, with cytoplasm filled with small yolk droplets, whereas stage IV and V contained large and fused yolk droplets respectively. Therefore, due to the presence of more lipid droplets, the intensity of staining of COX1 was too low to be detected. There are reports that the expression of COX1 varied with the stage of the oocyte and was more expressed in early and late previtellogenic stage oocytes [55,79]. Furthermore, the studies on the expression of COX1 in the ovary of the Ostrich revealed that the intensity of immunostaining of COX1 decreased with the transition of oocytes to the vitellogenic follicular stage [79]. Even though there was a decrease in the content of yolk droplets and an increase in the presence of vacuoles in the oocytes in the ovary of the ticks treated with flumethrin, the localization of COX1 could not be appreciated in the oocytes. Hence, a proper conclusion on the role of COX1 on the reproduction of ticks could not be made in the present study.

#### 2.3.2. Immunoperoxidase Staining with Anti-PGES Antibodies of the Ovarian Sections

The results of the immunoperoxidase staining with anti-PGES antibodies on the ovarian sections of ticks treated with methanol, flumethrin and terpenoid subfraction are depicted in Figure 3A–C, respectively. The ovarian tissues of the ticks treated with methanol showed localization of PGES around the cytoplasm of stage III oocytes. The ovary of the ticks treated with flumethrin and terpenoid subfraction showed no localization of the PGES enzyme. In the ovaries of the methanol-treated tick, the intensity of PGES was more around the cytoplasm near to nuclear region. The findings indicated that biosynthesis of prostaglandin occurred normally in the control group and the prostaglandins had an important role in the maturation of ovaries and oviposition. Moreover, the histological features of the ovary of the ticks treated with methanol (control) were comparable to that of a normal ovary [73]. Since the localization of the inducible COX2 was not studied, it was difficult to specify the type of PGES localized in the ovarian section. Therefore, it was not possible to conclude whether the PGES linked to COX2 was inhibited in the oocytes of the ovaries of *R. annulatus* ticks treated with flumethrin and terpenoid subfraction of *A. nilagirica*.

### 2.4. Quantification of PGE_2_ in the Engorged Tick Ovaries by Liquid Chromatography with Tandem Mass Spectrometry (LC-MS-MS)

#### 2.4.1. Response Linearity of PGE_2_

The working standards with ascending concentrations of PGE_2_ at 5, 10, 25, 50, 100, 250, 500 and 1000 pg/mL in methanol were analyzed using LC-MS-MS. The respective peaks with the area were calculated. The calibration curve of PGE_2_ concentration versus time was plotted and a linear regression was performed on the data set. The regression coefficient (R^2^) calculated was 0.999.

#### 2.4.2. Detection and Recovery of PGE_2_ from Tick Ovaries

Prostaglandin E_2_ was detected in methanol at the wavelength 196 nm with a retention time of three minutes. Recovery of PGE_2_ from tick ovaries was done using different concentrations viz., 1 ppm and 5 ppm spiked samples in methanol and the recovery % was 83.61 and 98.03 respectively.

#### 2.4.3. Detection of PGE_2_ in the Treated Tick Ovarian Samples

The ovarian samples from the ticks treated with methanol (control), flumethrin and terpenoid subfraction of *A. nilagirica* showed the detection of PGE_2_ in all the samples analyzed. In ticks treated with methanol, the PGE_2_ level detected was 24 pg/tick. The ovary of the ticks treated with flumethrin revealed the PGE_2_ levels below the detection limit (BDL). The ovary of ticks treated with terpenoid subfraction detected a mean PGE_2_ concentration of 4 pg/tick. The Table 3 depicts the mean concentration of PGE_2_ between the groups of the treated ticks.

Prostaglandin E_2_ is the most common prostanoid found in both mammalian and non-mammalian species with a variety of bioactivities and has various pathophysiological functions [80,81]. The role of PGE_2_ in controlling the reproduction was well established in insects, fish, crustaceans, arthropods and mammals as well as in humans [53,54,55,56,82]. In ticks, PGE_2_ is also involved in the regulation of ovarian maturation, oocyte development, egg-laying and associated behaviors in addition to the host-parasite relationship.

In the present study, the concentration of PGE_2_ quantified in the ovaries of untreated *R. annulatus* tick was the 24 pg/tick. The concentration of PGE_2_ levels in *B. microplus* ovaries was reported previously as 27 pg/tick [83]. The results indicated that there was not much variation in the levels of PGE_2_ in both species of ticks. On the other hand, the concentration of PGE_2_ was below the detectable limit in the ovaries of ticks treated with flumethrin. Moreover, the levels of PGE_2_ were reduced to 4 pg/tick in the ovaries of ticks treated with the terpenoid subfraction of *A. nilagirica*.

In another study conducted in our laboratory, it was observed that flumethrin at concentrations ranging from 30–100 ppm caused a complete inhibition of fecundity in ticks [84]. In the present study, the inhibition of fecundity of ticks caused by treatment with terpenoid subfraction of *A. nilagirica* was 62%, which was comparatively less than the effect caused by flumethrin. These findings indicated that there was a significant difference in the inhibition of PGE_2_ synthesis caused by the treatment with flumethrin and the terpenoid subfraction of *A. nilagirica* on ticks. Additionally, the few eggs laid by the ticks treated with the terpenoid subfraction could be correlated with the meagre quantity of PGE_2_ detected in the treated tick ovary. Correspondingly, the key enzyme PGES involved in the biosynthesis of PGE_2_ could not be detected in ovaries of the ticks treated with the flumethrin and terpenoid subfraction.

Many authors previously reported the gradual increase in the amount of PGE_2_, PGF_2α_ and several unsaturated fatty acids during vitellogenesis in the ovary of Florida freshwater crayfish [51,85]. The concentration of PGE_2_ in different stages of ovarian tissues of marine organisms varied from 20 ng/g tissue in the early stages to the highest levels in stage IV vitellogenic oocytes at 30 ng/tissue [86]. A very recent study demonstrated the implication of PGE_2_ in inducing vitellogenesis, ecdysteroidogenesis and MF synthesis in selected edible crustaceans [56]. The results from the present study thus provide strong evidence for PGE_2_ as an important molecule in signaling the oocyte maturation and in the laying of eggs in ticks. The data further support the concept that the reduced PGE_2_ levels in the ovaries of ticks treated with flumethrin and terpenoid subfraction could be one of the reasons for the complete/partial eclosion blocking effect elicited. Thus, like other non-insect arthropods, PGE_2_ is highly critical for reproduction in ticks.

### 2.5. Quantification of Vitellogenin (Vg) in the Hemolymph of R. annulatus Tick Using Indirect ELISA

The quantification of vitellogenin (Vg) in the hemolymph of ticks was done by the indirect ELISA method (Table 4). The optical density values in the methanol control, as well as the flumethrin and terpenoid treated groups, showed no significant difference. The anti-vitellogenin monoclonal antibody used in the present study was raised in mice. This could be the reason for the absence of any significant reactivity against the tick vitellogenin.

The vitellogenin and its receptor are the key components in the yolk formation in the developing oocytes of ticks [87]. In the flumethrin and terpenoid subfraction-treated tick ovaries, the yolk formation was disrupted, even though the level of vitellogenin in the hemolymph of these ticks were unaltered when compared to the untreated control ticks. Hence, the decrease in the content of yolk droplets in the ovary of treated ticks might be an indication of the inhibitory effect at the vitellogenin receptor level or at the level of the biosynthesis and release of vitellogenin. Even though a significant decrease in the PGE_2_ concentration was observed in treated tick ovaries, the present study, could not establish a relationship between the concentration of PGE_2_ and vitellogenesis.

In ticks, the process of vitellogenesis is induced by blood meal and is mediated by the hormone ecdysteroids [88]. Vitellogenin (Vg), is synthesized as a high molecular weight precursor in the fat body, midgut and ovaries of ticks. Vitellogenin released into the hemolymph is then taken up by the oocytes via receptor-mediated endocytosis. Once accumulated inside the oocyte yolk granules, it is converted to vitellin (Vn), which serves as the major source of nutrients for the development of the embryo [89]. A delayed ovary development and reduced fecundity were observed in *R. microplus* females where the vitellogenin receptor responsible for the uptake of Vg from the hemolymph into the oocytes was knocked down [89]. Additionally, avermectin analogue (MK-243) and cypermethrin inhibited the vitellogenesis and egg development in *Amblyomma hebraeum* Koch by interfering with the synthesis of vitellogenin [74] and the uptake of vitellogenin by the oocytes [89], respectively. Thus, it is presumed that there is a role for the vitellogenin in tick reproduction. However, in the present study we could not observe a significant variation in the vitellogenin concentration in the control and treated ticks.

Thus, the study demonstrated that the terpenoid subfraction isolated from the hexane fraction of ethanolic extract of *A. nilagirica* could disrupt the reproduction in *R. annulatus* ticks through the inhibition of PGE_2_ synthesis. The role of PGE_2_ on the ovarian maturation and oocyte development in ticks could be used for developing strategies for control of the ticks and tick-borne pathogens. The inhibitors of the PGES may prove beneficial in modulating the tick reproductive processes. The prostaglandin E_2_ receptors of ticks can be exploited as potential drug targets for tick control. The detection of PGES by immunohistochemistry and quantification of PGE_2_ by LC-MSMS can be employed as valuable tools for screening newer compounds for their eclosion-blocking acaricidal effects.

## 3. Materials and Methods

### 3.1. Chemicals

All the chemicals, solvents and reagents required for the research work were of a chromatographically pure grade and procured from M/s Merck India Ltd., Mumbai, India and M/s Sigma Aldrich India Ltd., Bangalore, India. The HRP-conjugated goat anti-rabbit IgG and rabbit polyclonal anti-COX and anti-PGES antibodies were purchased from M/s Santa Cruz Biotechnology, Inc., Dallas, TX, USA. The anti-vitellogenin (mouse monoclonal) antibody was obtained from M/s Abcam, Cambridge, UK. The DAB (3,3’Diaminobenzidine), TMB (3,3’,5,5’-tetramethylbenzidine) and OPD (Orthophenylenediamine) substrates were procured from M/s Thermo Fischer Scientific, Waltham, MA, USA. Water used in the study was purified in a Milli-Q^®^ system (M/s Millipore Bedford, MA, USA). The chemical, 9-oxo-11α, 15S-dihydroxy-prosta-5Z, 13E-dien-1-oic acid (PGE_2_) was procured from M/s Cayman Chemicals Co. (Ann Arbor, MI, USA). Flumethrin (with a purity of 99%) was procured from AccuStandard (New Haven, CT, USA). The LC MS grade solvents methanol, formic acid, *n*-hexane and ethyl acetate were procured from M/s Merck GmBH, Darmstadt, Germany.

### 3.2. Plant Materials

The aerial parts of the plant *A. nilagirica* (Clarke) Pamp. were collected from Kunnathidavaka village of Vythiri Taluk, Wayanad, Kerala, India before the flowering season. The collected plants were identified, authenticated by a botanist in the Department of Botany, University of Calicut. A voucher specimen of the plant was deposited in the University Herbarium CALI, the University of Calicut (Accession no. 88647). The plant materials were cleaned and kept for shade drying for two weeks at room temperature to remove moisture. The dried plant material was powdered using a temperature-controlled electrically operated plant sample grinder (M/s Rotek, Ernakulam, India) and was stored in an airtight container for extraction.

#### 3.2.1. Preparation of Terpenoid Subfraction

The powdered plant material (2 kg) was extracted using ethanol for eight to nine refluxes in Soxhlet’s apparatus and was reduced and concentrated under the pressure of 175 mbar at a temperature range of 65 °C in a rotary vacuum evaporator (M/s Buchi, Flawil, Switzerland). The ethanolic extract thus obtained was air-dried and was fractionated in a separatory funnel using hexane (polarity index: 0.1) to yield hexane fraction. The dried hexane fraction was subjected to isolation of terpenoids using the Herz-Hogenauer method [90]. Approximately 20 g of the dried hexane fraction was soaked overnight in dichloromethane (100 mL) and sonicated. The slurry product was then filtered and the green filtrate was evaporated to dryness in a rotary vacuum evaporator (M/s Buchi, Flawil, Switzerland). Furthermore, the residue obtained was dissolved in 95% ethanol (70 mL) and warmed at 70–80 °C to increase the solubility in a thermostatic water bath (M/s Thermo Scientific, Waltham, MA, USA). The aqueous solution (5%) of lead acetate was added to this in a dropwise manner (10 mL) for the precipitation of fatty acids, phenolics and chlorophyll. The precipitate was removed by filtration using a pad of silica gel (230–400 mesh, M/s Merck India Ltd., Mumbai, India). The filtrate was kept in the thermostatic water bath (M/s Thermo Scientific, Waltham, MA, USA) at 40–50 °C until the viscous mass remained. The subfraction obtained was stored at 4 °C and used for further chromatographic or spectral analysis and biological assays.

#### 3.2.2. Gas Chromatography–Mass Spectrometry Analysis of Terpenoid Subfraction

The terpenoid subfraction isolated from the hexane fraction of the ethanolic extract of *A. nilagirica* was analyzed by a single quadruple GCMS-QP-2010 Ultra, (Shimadzu Corporation, Kyoto, Japan) with an RXi 5 SILMS (Restek Corporation, Bellefonte, PA, USA) 30 m × 0.25 mm column having an internal diameter of 0.25 µm. Different concentrations of the terpenoid subfraction (10 ppm, 25 ppm, 50 ppm and 100 ppm) were prepared in one milliliter of hexane and from these working solutions, one microliter was injected into the GC-MS system in the split mode (split ratio 1:10). Helium was used as the carrier gas with a flow rate of 1 mL/min. The column oven temperature was maintained at 110 °C for two minutes. Then, it was programmed to 220 °C at a ramp rate of 10 °C/min. The final temperature of 280 °C was attained at a ramp rate of 5 °C/min and it was then held for nine minutes. The injector and detector temperatures were optimized at 250 and 280 °C, respectively. The MS operating parameters were as follows: ionization energy, 70 eV; ion source temperature, 280 °C; solvent delay, 1.0 min; and scan range, 100 to 350 *m*/*z*. The total run time was 36 min. The components were identified based on matching their retention indices available with the mass spectral library (Wiley 08, NIST 11).

### 3.3. Ticks

The fully engorged adult female *R. annulatus* ticks were collected from the naturally infested calves with a history of no prior exposure to any conventional acaricides. They were washed with distilled water and dried on an absorbent paper. These ticks were used for the adult immersion test (AIT).

The role of key enzymes in the prostanoid pathway and the prostaglandins in tick ovary were not understood in great detail. Hence, the collected ticks were also used for studying the localization of COX1 and PGES enzymes in the tick ovary by an immunoperoxidase assay. In addition, the levels of PGE_2_ in the engorged tick ovaries and vitellogenin concentration in the hemolymph were also evaluated. The synthetic pyrethroid flumethrin was used for comparison.

#### Adult Immersion Test (AIT)

The adult immersion test was performed based on Drummond et al. [91] The different concentrations of terpenoid subfraction (10% to 50%) were prepared in methanol. Four replicates, each with six ticks, were used for each concentration. The groups of six ticks selected randomly based on the size were weighed before the experiments and were immersed for two minutes in the respective dilution in a 50 mL beaker containing 10 mL extract. The ticks were recovered from the solution, dried using absorbent paper and placed in a separate plastic specimen tube (25 mm × 50 mm). The tubes were incubated at 28 ± 2 °C temperature and 80% relative humidity in a Biological Oxygen Demand (BOD) incubator. The adult tick mortality was observed up to the 15th day post-treatment. After oviposition, the eggs laid by the female ticks were collected and weighed. The eggs were kept under the same incubation conditions in a BOD incubator for the next 30 days.

The index of egg-laying (IF) and per cent inhibition of fecundity (InF) were calculated [92] as follows.
(1)$$\mathrm{Index}\;\mathrm{of}\;\mathrm{egg}\;\mathrm{laying}\;\left(\mathrm{IF}\right)=\frac{\mathrm{Weight}\;\mathrm{of}\;\mathrm{eggs}\;\mathrm{laid}\;\left(g\right)}{\mathrm{Weight}\;\mathrm{of}\;\mathrm{females}\;\left(g\right)}$$
(2)$$\mathrm{Percentage}\;\mathrm{inhibition}\;\mathrm{of}\;\mathrm{fecundity}\;\left(\mathrm{InF}\right)=\frac{\left[\mathrm{IE}\;\left(\mathrm{control}\;\mathrm{group}\right)-\mathrm{IE}\;\left(\mathrm{treated}\;\mathrm{group}\right)\right]\times 100}{\mathrm{IE}\;\left(\mathrm{control}\;\mathrm{group}\right)}$$
Percent inhibition of hatching of eggs.

The eggs laid by ticks of each tube were weighed and observed for the next 24 days in the BOD incubator for visual hatching of larvae.

### 3.4. Modulation of Prostaglandin-Mediated Pathway

#### 3.4.1. Calculation of Median Lethal Concentration (LC_50_)

The dose-response data were analyzed by the probit method [93] using the Graph Pad Prism 4 software (Graph Pad Software Inc., San Diego, CA, USA). The LC_50_ of the terpenoid subfraction was calculated by applying regression equation analysis to the probit-transformed data on the adult tick mortality and log concentration. The calculated LC_50_ value of terpenoid subfraction was used for assessing the modulation of the prostaglandin-mediated pathway. The synthetic pyrethroid flumethrin was used as a positive control. Flumethrin at 90 ppm was reported with 100 per cent inhibition of fecundity and nil hatching percentage [84].

#### 3.4.2. Localization of Cyclooxygenase1 (COX1) and Prostaglandin E Synthase (PGES) Enzymes in the Tick Ovary

Thirty fully engorged adult female ticks were used for studying the localization of COX1 and PGES enzymes in the tick ovary by immunoperoxidase assay. They were divided into three groups of ten ticks each and immersed for two minutes in a separate beaker containing 10 mL methanol 100% (Group I), flumethrin, 90 ppm (Group II), and terpenoid subfraction, 40% concentration (Group III) respectively. The ticks were then recovered from the respective solutions, dried on absorbent tissue paper towels and placed in separate plastic specimen tube (25 × 50 mm). They were kept at 28 ± 1 °C and 85 ± 5% relative humidity (RH) in a BOD incubator (M/s Kemi, Ernakulam, India). After 24 h, they were taken out from the incubator, dissected in 0.9% saline solution and the ovaries were removed under a stereo zoom microscope (M/s Labomed, Gurgaon, India).

The processing of the ovarian tissues of the tick was performed as per the method described by Sumpownon et al. [55] with minor modifications. The dissected tick ovarian tissues were collected and fixed in 4% paraformaldehyde in phosphate-buffered saline (PBS 0.1 M) and kept at 4 °C overnight. The tissues were processed in ascending concentrations of ethanol. Paraffin-embedded blocks were made and sections were taken at 3 µm thickness with low-profile microtome blades. These sections were deparaffinised by xylene. The descending concentrations of ethanol were used for the rehydration of tissues. The sections were then incubated in the dark in 3% H_2_O_2_ containing methanol at room temperature for 15 min to suppress the endogenous peroxidase. The sections were then washed with PBS and incubated in PBS containing 0.1% glycine for 15 min and washed again with PBS. The slides were incubated in blocking solution (PBS containing 4% bovine serum albumin, 5% normal goat serum and 0.4% Triton X-100) to arrest the nonspecific bindings for one hour at room temperature. The tissue sections were washed thrice with PBS. Then the sections were incubated overnight in rabbit polyclonal anti-COX1 and/or anti-PGES antibodies diluted at 1:200 in PBS and Tween 20 (PBST) separately. These sections were again incubated with Horseradish peroxidase (HRP) secondary antibody i.e, HRP-conjugated goat anti-rabbit IgG at 1:500 in PBST for two hours. The staining of the sections was developed using DAB substrate solution for 15 min. Some of these sections were counter-stained with hematoxylin and eosin for identification. Ultimately, the immunoperoxidase stained sections were examined under an inverted fluorescence microscope (Model: Axio Vert.A1 FL-LED, M/s Carl Zeiss Microscopy, Oberkochen, Germany) for the localization of enzymes, COX1 and PGES. The images were captured and analyzed using ProgRes^®^ CapturePro-Camera Control Software. (JENOPTIK/Optical Systems Version 2.8.8., Goeschwitzer Strasse, Germany).

#### 3.4.3. Determination of the Levels of PGE_2_ in Engorged Tick Ovaries by Liquid Chromatography with Tandem Mass Spectrometry (LC-MS-MS)

##### Instrumentation

A triple quadrupole mass spectrometer UHPLC-MS/MS 8030 (M/s Shimadzu Corporation, Tokyo, Japan) was used for the analysis of PGE_2_ from the tick ovaries. The high-pressure UHPLC (Nexera X_2_, M/s Shimadzu Corporation, Tokyo, Japan) was equipped with a DGU-20A5R degasser, LC-30AD quaternary gradient pump, a SIL 30AC autosampler, a column oven CTO-20AC, a diode array detector SPD-M20A, and a CBM-20A bus module. The chromatographic separation was performed on a UHPLC C_18_ (100 × 2.1 mm, 1.8 µm, Reverse phase) Phenomenex column. The Lab-solutions version software was used for the analysis of PGE_2_ in the tick ovary. The MS was operated in negative electrospray source ionization (ESI) and multiple reaction monitoring modes (MRM). The mobile phase used was acetonitrile and 0.1 per cent formic acid in the ratio of 80:20 at a flow rate of 0.15 mL and a run time of 5 min.

The standards of PGE_2_ were prepared from 1000 ppm (1000 µg/mL) stock in a suitable amount of methanol. The mass transition was scanned from 10 ppm concentration. Tandem MS was used for measuring selected ion through MRM mode. The product ions were selected for PGE_2_. The run time was 5 min. The mass transitions after MRM scan for PGE_2_ were *m*/*z* 351–271. The working standards were prepared from 5 pg/mLto 1000 pg/mL in a suitable quantity of methanol and stored at 4 °C for further LC-MS/MS analysis. For calculating the response linearity of PGE_2_, the ascending standards of PGE_2_ at 5, 10, 25, 50, 100, 250, 500 and 1000 pg/mL in methanol were analyzed using LC-MS/MS and peak areas were calculated. The calibration curve of PGE_2_ concentration versus area was plotted on a graph paper. The linear regression was performed on the data set and the regression coefficient (R^2^) was calculated.

##### Sample Preparation

Thirty fully engorged adult female ticks were used for the quantification of PGE_2_ in the tick ovary by LC-MS/MS. They were divided into three groups of ten ticks each and immersed for two minutes in a separate beaker containing 10 mL methanol (Group I), flumethrin, 90 ppm (Group II), and terpenoid subfraction, 40% concentration (Group III) respectively. The ticks were then recovered from the respective solutions, dried on absorbent tissue paper towels and placed in separate plastic specimen tubes (25 × 50 mm). They were kept in the BOD incubator. The ovaries of the fully engorged female ticks were dissected and weighed individually. The tissue was transferred separately to glass tubes and homogenized twice with 500 µL methanol. The supernatants were transferred to fresh glass tubes and were centrifuged at 1000× *g* for 10 min at 4 °C and then kept for drying. The extract was dissolved in 500 µL of methanol and then centrifuged at 1000× *g* for 15 min at 4 °C. This supernatant was transferred again to a fresh glass tube, dried and then was reconstituted with methanol and analyzed by LCMSMS. The area under the peak was measured to determine the concentration of PGE_2_ recovered from the ovarian tissues and compared using the associated RSD values.

#### 3.4.4. Indirect Enzyme-Linked Immunosorbent Assay (ELISA) for the Estimation of Vitellogenin Concentration in the Hemolymph of Tick

The iMark^TM^ microplate absorbance reader (M/s Bio-Rad, Hercules, CA, USA) with an absorbance range of 400–750 nm and the Microplate Manager^®^ 6 Software were used for the analysis. Thirty engorged female ticks were used for the quantification of vitellogenin (Vg) in the hemolymph of the tick by indirect enzyme-linked immunosorbent assay (ELISA). They were divided into three groups of ten ticks each and immersed for two minutes in separate beakers containing 10 mL methanol (Group I), flumethrin, 90 ppm (Group II), and terpenoid subfraction, 40% (Group III) respectively. Ticks were then incubated in BOD. After 24 h, the hemolymphs of ticks of all groups were collected using 10 µL microtips individually into 0.2 mL microtubes from the tick’s legs cut at the distal joint with the scissors under a stereo zoom microscope. The collected hemolymph was then diluted with 10 µL 1X PBS buffer. The concentration of protein in the hemolymph was measured in a spectrophotometer (M/s Thermo Scientific™ NanoDrop 2000, Waltham, MA, USA) at 280 nm absorbance. The concentration was adjusted to 2 µg/mL using a coating buffer (30 mM sodium carbonate and 75 mM sodium bicarbonate, pH 9.6). Fifty-six wells of ELISA plates were coated with the coating buffer containing hemolymph (100 µL) and incubated at 37 °C for 1 h or kept overnight at 4 °C. The plates were washed thrice with 1X PBS-Tween 20 (0.05%). Then, the plates were blocked with 1X PBS containing 3% skim milk powder (300 µL) and incubated at 4 °C overnight. After overnight incubation, the plates were again washed thrice with 1X PBS-Tween 20. Anti-vitellogenin antibody (diluted 1:300 in 1X PBS containing 1% skim milk powder) was added to each well (100 µL) and incubated at 37 °C for one hour. The plates were again washed 5 times with PBS-Tween 20. The Rabbit anti-mouse IgG-HRP conjugate (diluted 1:3000 in 1X PBS containing 1% skim milk powder) was added to each well (100 µL). The plates were then incubated at 37 °C for one hour and again washed five times with PBS-Tween 20. Orthophenylenediamine (OPD) was dissolved 1 mg/mL concentration in the substrate buffer (0.1 M citric acid and 0.2 M disodium hydrogen phosphate). The freshly prepared substrate solution was taken in a trough and H_2_O_2_ (3%) was added. The final substrate solution (100 µL) was added to each well and incubated for 10 min in the dark. After incubation, 50µL of 3N HCl was added to each well to stop the reaction. The absorbance was read at 450 nm in an ELISA plate reader (M/s Bio-Rad, Hercules, CA, USA).

### 3.5. Statistical Analysis

The statistical analysis was done according to standard procedure as described by [94] Snecdecor and Cochran. All the data were expressed as Mean ± SEM. Treatment groups were compared using one-way ANOVA in SPSS Software Version 21 (SPSS Inc., Chicago, IL, USA). For the calculation of LC_50_ values, the probit analysis method and GraphPad Prism4 software were used for the analysis of the data. Post-hoc analysis was done using Duncan’s multiple range test. Value of *p* less than 0.05 was considered as significant.

## 4. Conclusions

The present study revealed that the plant *A. nilagirica* is a novel source for development as a botanical acaricide. The flumethrin and the terpenoid subfraction of *A. nilagirica* has shown the concentration-dependent effects on the mortality, inhibition of fecundity and eclosion-blocking effects against *R. annulatus* ticks. The histological analysis of sections of the ovary revealed abnormal morphological and structural changes with increased vacuolation and diminution in the content of yolk granules in vitellogenic oocytes. It was found that the flumethrin and terpenoid subfraction of *A. nilagirica* caused a remarkable reduction in the concentration of PGE_2_ in the ovary of treated ticks. Moreover, they prevented the immunoreactivity of PGES in the oocytes of the ovaries of *R. annulatus* ticks. The study thus provided strong evidence to suggest that the prostaglandin, PGE_2_ and its biosynthesis, are very much important for signaling the ovarian maturation and oviposition in ticks. This may provide a potential target for developing newer strategies for the control of ticks and tick-borne pathogens. The PGE_2_ and the key enzyme PGES can be considered as potential biomarkers for screening new drug entities for their acaricidal properties.

## Figures and Tables

**Figure 1 molecules-26-04905-f001:**
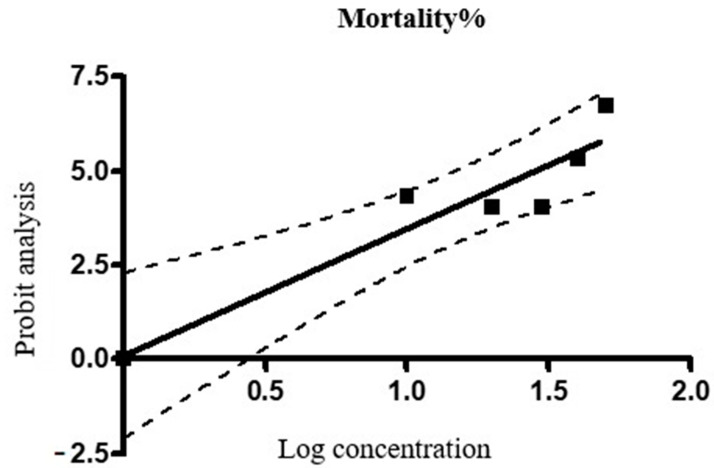
Dose-response curve of per cent adult tick mortality.

**Figure 2 molecules-26-04905-f002:**
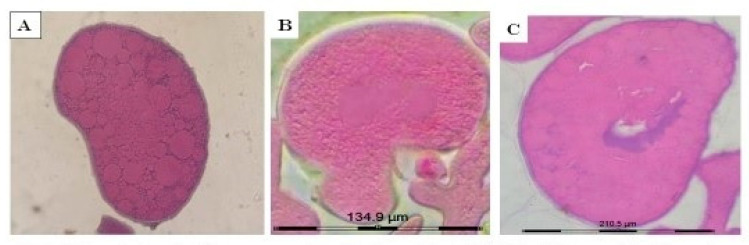
Microphotograph of immunoperoxidase labelling with anti-COX1 in *R. annulatus* tick ovaries. (**A**): Oocyte of ovary of control tick showing intact membrane, yolk droplets and cytoplasm; with no localization of COX-1enzyme; (**B**): Oocyte of ovary of ticks treated with flumethrin (90 ppm) showing irregular shaped oocyte and disrupted yolk droplets with no localization of COX-1enzyme; (**C**): Oocyte of ovary of tick treated with terpenoid subfraction of *A. nilagirica* (40%) showing degeneration of nuclear region and disrupted yolk granules with no localization of COX-1 enzyme.

**Figure 3 molecules-26-04905-f003:**
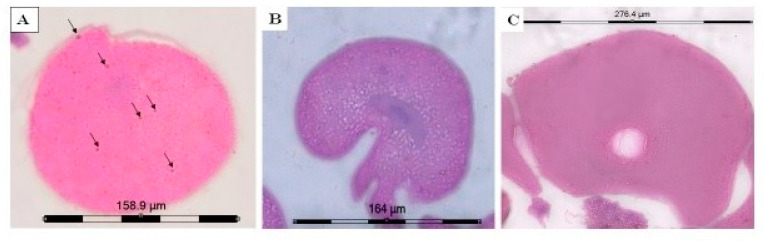
Microphotograph of immunoperoxidase labelling with anti-PGES in *R. annulatus* tick ovaries. (**A**): Oocyte of ovary of control tick showing intact membrane, yolk droplets and cytoplasm; Arrows indicates the presence of PGES enzymes; (**B**): Oocyte of ovary of tick treated with flumethrin (90 ppm) showing irregular shaped oocyte and disrupted yolk droplets with no localization of PGES; (**C**): Oocyte of ovary of tick treated with terpenoid subfraction of *A. nilagirica* (40%) showing degeneration of nuclear region and disrupted yolk granules with no localization of PGES.

**Table 1 molecules-26-04905-t001:** GC-MS analysis of terpenoid subfraction of hexane fraction of ethanolic extract of *A. nilagirica*.

Peak No.	Name of Compound	RT	Area %
1	Neoisothujyl alcohol or neoisothujol	4.541	4.19
2	Ledol	10.139	1.11
3	Cycloisolongifolene, 8,9-dehydro-	10.34	2.05
4	beta-eudesmol	10.71	2.72
5	Longipinocarveol, trans-	11.624	1.7
6	Z,Z-2,15-Octadecedien-1-ol acetate	12.7	2.29
7	1,2-Benzenedicarboxylic acid, bis(2-methylpropyl) ester (diisobutyl phthalate)	12.945	2.03
8	3-Thujanone	11.309	0.54
9	Isothujol	12.135	1.83
10	Isothujol	13.01	7.46
11	trans-3(10)-Caren-2-ol	16.215	0.32
12	Spathulenol(1*H*-Cycloprop[e]azulen-7-ol)	19.202	0.99
13	Caryophyllene oxide	19.845	0.84
14	Ledol	21.047	0.44
15	Tetracyclo [6.3.2.0(2,5).0(1,8)]tridecan-9-ol,4,4-di methyl-	21.724	1.99
16	4-epi-cubedol	21.79	0.7
17	β-Eudesmol	21.885	7.05
18	Caryophyllene oxide	21.955	0.96
19	tau.-Cadinol	22.125	0.45
20	4,6,6-Trimethyl-2-(3-methylbuta-1,3-dienyl)-3-oxatricyclo[5.1.0.0(2,4)]octane	22.292	0.54
21	9,10-Dimethyltricyclo[4.2.1.1(2,5)]decane-9,10-diol	22.545	1.03
22	(−)-Spathulenol	23.11	1.78
23	Nootkaton-11,12-epoxide	23.415	1.09
24	Longifolene-(I2)-epoxide-(1)	24.129	1.55
25	1,2-Benzenedicarboxylic acid, bis(2-methylpropyl) ester (diisobutyl phthalate)	24.172	4.87
26	alpha-Copaen-11-ol(Tricyclo[4.4.0.0(2,7)]dec-8-ene-3-methanol)	24.293	0.74
27	Phytol (3,7,11,15-Tetramethyl-2-hexadecen-1-ol)	24.409	1.37
28	Platambin	24.546	1.38
29	Alloaromadendrene oxide-(1)	24.604	0.4
30	Ethyl palmitate (Hexadecanoic acid, ethyl ester)	25.438	4.79
31	Phytol	26.854	4.36

**Table 2 molecules-26-04905-t002:** Effect of terpenoid subfraction of hexane fraction of ethanolic extract of *A. nilagirica* against *R. annulatus* at different concentration.

Sl.	Acaricide	Mean Ticks Weight per Replicate ± SEM	Mean (%) Adult Mortality within 15 Days ± SEM	Mean Eggs Mass per Replicate ± SEM	Index of Fecundity ± SEM	Percentage Inhibition of Fecundity (%)	Hatching (%)
No.		(g)		(g)			(Visual)
1	Methanol (control)	0.94 ± 0.02 ^a^	0 ± 0 ^a^	0.32 ± 0.06 ^a^	0.35 ± 0.06 ^a^	0	100%
2	10%	0.98 ± 0.00 ^a^	25 ± 8.33 ^c^	0.32 ± 0.03 ^a^	0.33 ± 0.03 ^a^	5.5	50%
3	20%	0.95 ± 0.00 ^a^	16.66 ± 6.80 ^d^	0.3 ± 0.05 ^a^	0.31 ± 0.06 ^a^	4.1	50%
4	30%	0.97 ± 0.01 ^a^	16.67 ± 11.79 ^d^	0.28 ± 0.02 ^a^	0.29 ± 0.02 ^b^	11.82	25%
5	40%	0.94 ± 0.02 ^a^	66.66 ± 6.80 ^b^	0.11 ± 0.01 ^b^	0.12 ± 0.01 ^c^	62.85	5%
6	50%	0.93 ± 0.02 ^a^	95.83 ± 4.17 ^a^	0.02 ± 0.01 ^b^	0.02 ± 0.01 ^d^	94.19	0

Number of replicates = 4, Number of ticks per replicate = 6. Values are Mean ± SEM, means bearing different superscripts ^a^, ^b^ or ^c^ (*p* < 0.05), indicate significant difference when compared with the control and recommended concentration of terpenoid subfraction.

**Table 3 molecules-26-04905-t003:** Mean concentration of PGE_2_ detected in the ovaries of ticks treated with methanol, flumethrin and terpenoid subfraction of hexane fraction of ethanolic extract of *A. nilagirica*.

Treated Groups	PGE_2_ Concentration (pg/tick) (*n* = 6; Mean ± SE)
Methanol (control)	24.1603 ± 2.484 ^a^
Flumethrin	BDL
Terpenoid subfraction	4.5448 ± 0.074 ^b^

BDL: Below detection limit; Values are Mean ± SEM and different subscripts ^a^, ^b^ indicates significant difference when compared with control group (*p* < 0.05).

**Table 4 molecules-26-04905-t004:** Optical density of vitellogenin (Vg) concentration in the hemolymph of treated ticks.

Treated Group	Optical Density Values (*n* = 6; Mean ± SEM)
Methanol (Control)	0.1417 ± 0.0050 ^a^
Flumethrin	0.1503 ± 0.0087 ^a^
Terpenoid subfraction	0.1525 ± 0.0037 ^a^

Values are Mean ± SEM and ‘^a^’ indicates no significant difference when compared with control group (*p* < 0.05).

## Data Availability

The data presented in this study are available on request from the corresponding author.

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
