# Peer review of "Modulation of the PGE2-Mediated Pathway in the Eclosion Blocking Effect of Flumethrin and Terpenoid Subfraction Isolated from Artemesia nilagirica in Rhipicephalus annulatus"

_molecules, 2021, doi:10.3390/molecules26164905_

Round 1
Reviewer 1 Report
The manuscript presented by the authors describes a new potential tool for tick control. The results support the authors' claims but there is no discussion on the application potential of the tool. The authors always remain on the technical side. How is their tool innovative? what advantages does it offer?
However, I believe that the authors need to make some changes before the manuscript can be accepted for publication.
1) The English draft of the manuscript must be strongly revised. There are often very complex sentences that impair understanding of the text.
2) the abstract should be edited. It is too long and goes into too much technical detail. In my opinion, a more functional abstract should better summarize the methods and results and emphasize the purpose and innovation of the work. The importance of the results and their use in an application perspective is completely missing.
3) In the Results section, I would suggest that the author's review Table 2. The footnotes are not very clear. The authors report in the table six samples (n = 6) but in the materials and methods, they speak of 4 replicates of 6 samples each? The mean is calculated between which samples?
4) The authors discuss the results in a very technical way. Totally missing a discussion of the potential of their results in terms of benefits and innovation in the Results and Discussion section or Conclusions (better here in my opinion).
5) In the Materials and Methods section: What kind of objective did the authors use for the localization of COX1 and PGES?
For these reasons, I suggest re-evaluating the manuscript for publication after these changes.
Author Response
Reply to comments of Reviewer 1
- The manuscript presented by the authors describes a new potential tool for tick control. The results support the authors' claims but there is no discussion on the application potential of the tool. The authors always remain on the technical side. How is their tool innovative? what advantages does it offer?
Response: Included in the discussion and conclusion as suggested
- The English draft of the manuscript must be strongly revised. There are often very complex sentences that impair understanding of the text.
Response: We have thoroughly edited the manuscript to correct the typos, mistakes and preventing the complex sentences.
2) The abstract should be edited. It is too long and goes into too much technical detail. In my opinion, a more functional abstract should better summarize the methods and results and emphasize the purpose and innovation of the work. The importance of the results and their use in an application perspective is completely missing.
Response: The abstract is revised thoroughly as suggested.
3) In the Results section, I would suggest that the author's review Table 2. The footnotes are not very clear. The authors report in the table six samples (n = 6) but in the materials and methods, they speak of 4 replicates of 6 samples each? The mean is calculated between which samples?
Response: Footnotes are revised. Mean is calculated for the replicates.
4) The authors discuss the results in a very technical way. Totally missing a discussion of the potential of their results in terms of benefits and innovation in the Results and Discussion section or Conclusions (better here in my opinion).
Response: Added in the discussion and conclusion
5) In the Materials and Methods section: What kind of objective did the authors use for the localization of COX1 and PGES?
Response: Under the heading Materials and methods, 3.3 ticks, we have included
“The role of key enzymes in the prostanoid pathway and the prostaglandins in tick ovary were not understood in great detail. Hence, the collected ticks were also used for studying the localization of COX1 and PGES enzymes in the tick ovary by immunoperoxidase assay.In addition, the levels of PGE2 in the engorged tick ovaries and vitellogenin concentration in the haemolymph were also evaluated.The synthetic pyrethroid, flumethrin was used for comparison.”
Reviewer 2 Report
The manuscript is interesting and the subject is worth of the investigation. The paper is quite well and clearly written. I have only few suggestions mostly editorial nature.
Line 33: „Liquid Chromatography’ – unnecessary capital letters (the same remark for line 271)
Line 118-120: Does percentage values mean the content in hexane fraction? They differ from values given in Table 1.
Check carefully Table 1: peak 3 „8,9-dehydro” – this is probably the first part name of compound no 4; peaks no 9 and 10 have the same names, peak 15 – is it full name?
Figure 1 is not informative. The quality should be improved. Use the numbers for peaks (according to Table 1) instead of retention times. Alternatively, the chromatogram could be move to Supplementary material.
Line 130: It should be: [60-62].
2.1. Section: some detail on the diversity of composition comparing to literature data should be added.
Line 134: “The variation could also be noticed in the composition between the essential oil” –, why was essential oil composition discussed? Solvent extract was the subject of interest in presented study.
Table 2: too many significant digits for % values
Figure 3: A, B, C – should be clearly described in Figure legend (the same remark for Fig. 4).
Table 3 is unnecessary in main body of the manuscript. Move it to Supplementary material (the same remark for Fig.5 and 6)
Line 519: It should be: “1000 ug/L”
Line 523: It should be: “from 5 pg/mL to 1000 pg/mL”
Line 527: It should be: “peak areas were calculated”.
Author Response
Reply to comments of Reviewer II
- Line 33: „Liquid Chromatography’ – unnecessary capital letters (the same remark for line 271)
Response: Correction done as suggested by the learned reviewer
2a. Line 118-120: Does percentage values mean the content in hexane fraction? They differ from values given in Table 1.
Response: It is relative concentration of a particular compound in the hexane fraction.
We added the % area of different compounds (For eg. 4.19+1.11+2.05+-------------+1.03+1.78) =63.56.
For neoisothujol the relative concentration is 4.19/63.56X100=6.59.
For Isothujol, 1.83+7.46=9.29. The relative concentration is (9.29/63.56)100=14.6 %
2b. Check carefully Table 1: peak 3 „8,9-dehydro” – this is probably the first part name of compound no 4; peaks no 9 and 10 have the same names, peak 15 – is it full name?
Response: Peak 3 RT 10.34 -name is Cycloisolongifolene, 8,9-dehydro- PubChem CID is 594593
Peak no 9 and no 10 gave the same names, however they were shown as two different peaks and NIST library search, reveal the same name for the peaks. It may be because, they may be isomers.
Peak 15 Yes. It is full name of single compound. Tetracyclo[6.3.2.0(2,5).0(1,8)]tridecan-9-ol,4,4-di methyl-.Pubchem CID is 585744
- Figure 1 is not informative. The quality should be improved. Use the numbers for peaks (according to Table 1) instead of retention times. Alternatively, the chromatogram could be move to Supplementary material.
Response: Removed figure 1
- Line 130: It should be: [60-62].
Response: corrected
- 2.1. Section: some detail on the diversity of composition comparing to literature data should be added.
Line 134: “The variation could also be noticed in the composition between the essential oil” –, why was essential oil composition discussed? Solvent extract was the subject of interest in presented study.
Response: Most of the previous literatures were on the contents of essential oil of Artemesia nilagirica. We were not getting any references related to the composition of solvent extracts of the plant. In order to compare the diversity in composition of the plant, we were forced to discuss the data on essential oil.
- Table 2: too many significant digits for % values
Response: corrected in the table as suggested
- Figure 3: A, B, C – should be clearly described in Figure legend (the same remark for Fig. 4).
Response: corrected
- Table 3 is unnecessary in main body of the manuscript. Move it to Supplementary material (the same remark for Fig.5 and 6)
Response: Table 3, Fig.5 and 6 were removed.
- Line 519: It should be: “1000 ug/L”
Response: corrected as 1000µg/mL
- Line 523: It should be: “from 5 pg/mL to 1000 pg/mL”
Response: corrected
- Line 527: It should be: “peak areas were calculated”.
Response: corrected